# Associations between Family-Based Stress and Dietary Inflammatory Potential among Families with Preschool-Aged Children

**DOI:** 10.3390/nu13051464

**Published:** 2021-04-26

**Authors:** Valerie Hruska, Nitin Shivappa, James R. Hébert, Alison M. Duncan, Jess Haines, David W. L. Ma

**Affiliations:** 1Department of Human Health and Nutritional Sciences, University of Guelph, Guelph, ON N1G 2W1, Canada; vhruska@uoguelph.ca (V.H.); amduncan@uoguelph.ca (A.M.D.); 2Cancer Prevention and Control Program, University of South Carolina, Columbia, SC 29208, USA; shivappa@email.sc.edu (N.S.); JHEBERT@mailbox.sc.edu (J.R.H.); 3Department of Epidemiology and Biostatistics, Arnold School of Public Health, University of South Carolina, Columbia, SC 29208, USA; 4Department of Nutrition, Connecting Health Innovations LLC, Columbia, SC 29201, USA; 5Department of Family Relations and Applied Nutrition, University of Guelph, Guelph, ON N1G 2W1, Canada; jhaines@uoguelph.ca

**Keywords:** dietary inflammatory index, inflammation, stress, mental health, family, health behaviour, diet quality, disease prevention

## Abstract

Chronic stress is known to influence dietary choices, and stressed families often report poorer diet quality; however, little is known about how family-based stress is linked with dietary patterns that promote inflammation. This study investigated associations between family-based stress and the inflammatory potential of the diet among preschool-aged children and their parents. Parents (*n* = 212 mothers, *n* = 146 fathers) and children (*n* = 130 girls, *n* = 123 boys; aged 18 months to 5 years) from 241 families participating in the Guelph Family Health Study were included in the analyses. Parents reported levels of parenting distress, depressive symptoms, household chaos, and family functioning. The inflammatory potential of parents’ and children’s diets was quantified using the Dietary Inflammatory Index (DII^®^), adjusted for total energy intake (i.e., the E-DII^TM^). E-DII scores were regressed onto family stress using generalized estimating equations to account for shared variance among family clusters. Compared to those in homes with low chaos, parents in chaotic homes had significantly more proinflammatory dietary profiles (*β* = 0.973; 95% CI: 0.321, 1.624, *p* = 0.003). Similarly, compared to those in well-functioning families, parents in dysfunctional families had significantly more proinflammatory dietary profiles (*β* = 0.967; 95% CI: 0.173, 1.761, *p* = 0.02). No significant associations were found between parents’ E-DII scores and parenting distress or depressive symptoms, nor were any associations found for children’s E-DII scores. Results were not found to differ between males and females. Parents in chaotic or dysfunctional family environments may be at increased risk of chronic disease due to proinflammatory dietary profiles. Children’s dietary inflammatory profiles were not directly associated with family stress; however, indirect connections through family food-related behaviours may exist. Future research should prioritize elucidating these mechanisms.

## 1. Introduction

Mental and physical health are often considered to be independent phenomena; however, growing evidence supports the many degrees to which these domains intersect and interact to determine overall health status. Dietary intake may be a key factor influencing the ways in which mental well-being and metabolic health are connected [1]. Accumulating evidence indicates that dietary patterns may play neuroactive roles, meaning that they influence neurological function and, by extension, cognition and mood. The Mediterranean diet, for example, has long been lauded for cardiovascular benefits, and more recently has been associated with lower levels of psychological distress [2], better cognitive function [3], and heightened self-perceived mental health status [4]. Likewise, dietary patterns consisting mostly of fruit, vegetables, lean proteins, and low intake of refined grains are associated with lower rates of major depression and anxiety [5,6]. Conversely, low-quality and Westernized dietary patterns higher in saturated and *trans* fats, refined grains, processed sugar, and fast food have been associated with poorer mental health statuses [7,8].

Neuroinflammation and oxidative stress are implicated in the pathogenesis of several mental health disorders, including depression and anxiety [9,10]. Suggested mechanisms through which inflammation affects mental health outcomes include decreased neurotransmitter production, impaired hormone synthesis, and dysregulated synaptic plasticity [11,12]. Diet may further contribute to these dysregulations, as evident in cross-sectional research showing that proinflammatory dietary patterns are associated with increased risk of mental health disorders [13]. However, within this emerging branch of research, less is known about how subclinical psychological distress such as chronic stress may contribute to these phenomena. 

Families with young children are a group of particular interest for several key reasons. First, parents of young children are likely to experience high levels of stress: reduced sleep, additional responsibilities, children’s dependency on parents to meet basic needs, managing children’s challenging behaviours, balancing demands of parenting with work or other life domains, and financial costs of children all contribute to demands on parents’ time and resources [14,15]. Abundant evidence shows that perceived stress promotes the consumption of sugary and fatty foods [16,17], often displacing more healthful options such as fruits, vegetables, and other foods rich in dietary fibre [18]. Research has shown that family stress is positively associated with parents’ dietary fat intake [19] and indices of adiposity [20], and negatively associated with family meal preparation and meal healthfulness [21,22]. As a modifiable source of inflammation, further research is needed to better understand how dietary patterns and food-related behaviours can be used to protect the health of stress-vulnerable individuals. 

Parents serve as gatekeepers to their children’s health practices; this enables stress to act multigenerationally to influence children’s health outcomes and behaviours. Parent stress has been shown to be associated with children’s poorer overall diet quality [23], adiposity [23,24], and health-related behaviours such as screen time and active play [25]. Additional importance is placed on families as a study group because young children are in a critical window of development during which these patterns may be solidified into trends that persist to adulthood [26]. Family health optimization begins with a better understanding of both parent and child perspectives. 

This study aimed to explore whether family-based stress is associated with inflammatory dietary patterns among families with young children. Given evidence demonstrating a sex-dependent influence of stress on dietary behaviours [17], this study also examined whether any associations between stress and inflammatory dietary patterns differed between males and females. 

## 2. Materials and Methods

### 2.1. Study Participants

Data were collected from families participating in the Guelph Family Health Study, a family-based longitudinal study investigating early life risk factors for obesity and chronic disease and family-based approaches for health promotion [27]. To be eligible, families had to have had at least one child aged 18 months to 5 years at the time of recruitment; reside in Wellington County, Ontario, Canada, with no plans to move in the following year; and have a parent who could respond to questionnaires in English. The study was approved by the University of Guelph Research Ethics Board (REB14AP008). Data collection took place between April 2017 and March 2020. 

### 2.2. Data Collection

#### 2.2.1. Family Stress Measures

Several measures, assessed via online surveys, were used to provide a composite picture of stress within the home. First, stress relating specifically to the role of being a parent was evaluated using the 12-question Parental Distress subscale of the Parenting Stress Index Short Form [28]. On a 5-point Likert scale from 1 (strongly disagree) to 5 (strongly agree), participants responded to statements such as “I often have the feeling that I cannot handle things very well”, “I feel trapped by my responsibilities as a parent”, and “Having a child has caused more problems than I expected in my relationship with my spouse (or male/female friend)”. A total score out of 60 was obtained by summing the responses with higher scores indicating greater parental distress. The midpoint of the scale was used to create “high distress” (≥36 points) and “low distress” groups. Standardized Cronbach’s alpha for this survey in our sample was 0.80.

Parental depressive symptoms were examined using a 10-item short form of the Centre for Epidemiologic Studies Depression scale [29]. On a 4-point scale from 0 (less than one day last week) to 3 (5–7 days), participants responded to statements such as “Everything I did was an effort” and “My sleep was restless”. A final score was calculated as the sum of responses. As per clinical guidelines [30], scores of 10 points or higher were considered “high depressive symptoms”, and scores less than 10 points were grouped as “low depressive symptoms”. Standardized Cronbach’s alpha for this survey in our sample was 0.81.

Home environment chaos was evaluated using the Confusion, Hubbub, and Order Scale (CHAOS), a quantification of a home’s disorganization, noisiness, and busyness [31]. Because this scale evaluates the home environment as a whole, only one parent was asked to respond on behalf of the family; the GFHS defaults to the first parent to contact the study regarding enrolment as the primary contact person for the family, of whom, 88% were mothers. For CHAOS, the primary parent responded to 15 statements such as “We almost always seemed to be rushed” and “It’s a real zoo in our home” on a 4-point Likert scale from 1 (very much like your own home) to 4 (not at all like your own home). A total score out of 60 was obtained by summing the responses, where higher scores indicate greater home chaos; this total score was applied to all members of the family. The midpoint of the scale was used to create “high chaos” (≥37.5 points) and “low chaos” groups. Standardized Cronbach’s alpha for this survey in our sample was 0.78.

Family functioning was examined using the 12-item general functioning subscale of the McMaster Family Assessment Device [32,33]. This scale consists of items such as “Planning family activities is difficult because we misunderstand each other” (reverse-scored) and “Making decisions is a problem for our family” (reverse-scored); participants responded on a 4-point Likert scale from 1 (strongly agree) to 4 (strongly disagree). The final score was calculated as a mean of the items, with greater scores indicating greater family dysfunction. A cut-point of ≥2.17 points was used to classify “high family dysfunction” because it has previously been shown to discriminate between healthy and unhealthy functioning in families with young children [32,33]. Similar to CHAOS, the primary parent was asked to respond on behalf of the family. Standardized Cronbach’s alpha for this survey in our sample was 0.88. 

#### 2.2.2. Dietary Measures 

Dietary assessment was conducted by the parents for themselves and on behalf of their children using the National Cancer Institute’s Automated Self-Administered 24-Hour (ASA24) Dietary Assessment Tool 2016-Canadian version [34], an online 24 h recall program validated for use among adults [35] and children [36]. The program is based on a modified version of the United States Department of Agriculture’s Automated Multiple-Pass Method for interviewer-led 24 h recalls. Food and drink terms were derived from the National Health and Nutrition Examination Survey. ASA24 includes a bank of food images to assist respondents with portion size estimation, and the database includes nutrient content of nearly 4000 foods, beverages, and supplements. Energy intake and nutrient intakes were quantified by the ASA24 program. 

The inflammatory potential of the diet was examined using the Dietary Inflammatory Index (DII^®^). The development and validation of the DII has been explained in detail elsewhere [37,38]. Briefly, the DII is a measure of the inflammatory potential of the diet with respect to circulating inflammatory markers such as c-reactive protein; tumor necrosis factor α; and interleukins 1β, 4, 6, and 10. Thus, the DII estimates the influence of diet on systemic inflammation. Inflammatory effect scores (the pro- or anti-inflammatory effect magnitude of a nutrient) were calculated using meta-analysis of nearly 2000 peer-reviewed articles. A total of 45 dietary parameters (including whole foods, spices, micro- and macronutrients, vitamins, trace minerals, and various other bioactive compounds) were identified as significantly impacting systemic inflammation based on a review of over 1900 peer-reviewed articles linking diet to key inflammatory biomarkers [37]. Parent DII scores were calculated based on intake quantities of 28 nutrients obtained from ASA24: total fat, saturated fat, monounsaturated fat, polyunsaturated fat, omega-3 polyunsaturated fatty acids, omega-6 fatty acids, trans-fat, carbohydrates, fibre, protein, cholesterol, iron, vitamin A, vitamin C, vitamin D, vitamin E, niacin, thiamin, riboflavin, vitamin B_6_, vitamin B_12_, folic acid, magnesium, zinc, selenium, alcohol, and caffeine. Participants’ DII score was calculated as the sum of the nutrient parameter intakes (standardized to global intake means as a *z*-score) as a function of the nutrient’s inflammatory effect score. The standardization of nutrients to global intake amounts better reflects the diversity of typical dietary patterns across the world. Scores range from −9 (most anti-inflammatory) to +8 (most proinflammatory). To control for the effect of total caloric intake, energy-adjusted DII (E-DII^TM^) scores were adjusted for energy by calculating nutrient amounts per 1000 calories of food consumed and then compared with the energy-standardized world databases [37], in a manner identical to that used for the DII. The exact scoring protocol has been described in detail elsewhere [37,38]. The DII and E-DII are scored similarly and scaled identically; thus, scores are comparable across studies.

Children’s DII scores were calculated using the adapted children’s protocol, the Children’s Dietary Inflammatory Index^TM^ (C-DII^TM^) [39]. The C-DII differs from the adult DII in that calculations are based on children’s intake databases, but the development protocols are otherwise virtually identical and described in detail elsewhere [39]. C-DII scores were similarly adjusted for total energy intake. C-DII scores range from −8 (most anti-inflammatory) to +8 (most proinflammatory). As with the DII and E-DII, the C-DII scores are comparable to the other studies.

### 2.3. Covariates

Parent models were adjusted for sex, age, body mass index (BMI), self-reported annual household income (5 categories: <$30,000; $30,000–59,999; $60,000–99,999; $100,000+; and “did not disclose”), self-identified ethnicity (6 categories: Northeast or Southeast Asian; South Asian; White; mixed ethnicity; other ethnicity; and “did not disclose”), and education (3 categories: no postsecondary degree; postsecondary graduate; postgraduate training). Household income, ethnicity, and education information was obtained via an online questionnaire. If parents from the same household reported different pooled annual incomes, the response from Parent 1 was used for both. Parents’ ages were calculated in years from their birthdate and the date that the ASA24 was completed.

Children’s models were adjusted for BMI z-score, sex, and age (in years) at the time that the ASA24 was completed.

#### BMI and BMI Z-Score Calculations

BMI was calculated using researcher-measured height and body mass at the University of Guelph Body Composition Lab. Height was measured while in a standing position at the apex of an inhaling breath to the nearest 0.1 cm using a wall-mounted stadiometer (Medical Scales and Measuring Devices; Seca Corp., Ontario, CA, USA). Two height measures were taken and averaged to give a final data point. If measures differed by more than 0.5 cm, a third measure was taken, and the final data point was an average of the two closest measures. For two child participants, a second measure was refused and so height from only the first measure was used. Measures from children who dissented or could not remain still were not used.

Participant body mass was measured in kilograms using a digital scale (Cosmed Inc., Concord, CA, USA), which was quality-control tested using two standardized 10 kg weights prior to participant measurements. Body mass was measured only once due to the precision and accuracy of the scale [40]. For 20 children, body mass was measured while in their parent’s arms and parent weight was subtracted to yield the child’s body weight. BMI values were calculated for parents as weight (kg) divided by height (m^2^). For children, BMI-z scores were calculated as per the WHO Child Growth Standards using the statistical software R package “zscorer” version 0.3.1 [41], which includes adjustment for child age and sex. 

### 2.4. Data Exclusions and Statistical Analyses

A total of 246 families (427 parents, 322 children) were enrolled in the GFHS cohort; of these, 37 parent and 30 child participants were excluded from these analyses due to missing dietary data. An additional 15 child participants were excluded from these analyses because they were breastfed and intake amounts could not be verified. Further, four parent and three child participants were excluded from the analyses because their total energy intakes were identified as mathematical outliers (>1.5 times the interquartile range below the 25th or above the 75th percentile) and were implausible based on a detailed examination of the ASA24. Finally, 28 parent and 21 child participants were excluded due to missing or incalculable BMI or BMI *z*-score data, used as covariates in these analyses, including 17 pregnant parent participants for whom BMI values are not validated. This yielded a final analytic sample of 358 parent participants (212 mothers, 146 fathers) and 253 child participants (130 females, 123 males) from 241 families.

Additional exclusions were made for participants who responded “I am not comfortable answering this question” to a stress-scale item because of which a final score could not be calculated. These exclusions were made on a scale-by-scale basis; no participants were missing responses for multiple scales, and so they were retained in the completed scale samples. The PSI scoring protocol allows for up to one missing response to be replaced by the mean of the other 11 responses [28]; this was done for *n* = 1 participant, who was retained in all analytic samples. No similar imputations were recommended by the other scale scoring protocols. For depressive symptoms, two participant records were incomplete; sample size for depression analyses was *n* = 356 parent participants. For CHAOS, records for five families (8 parents, 3 children) were incomplete; the sample size for CHAOS analyses was 350 parent and 250 child participants. For family functioning, records were incomplete for three families (3 parents, 4 children); the sample size for family functioning analyses was 355 parent and 250 child participants. 

Statistical analyses were conducted using SAS University Edition version 3.8 [42]. Linear regression coefficient estimates (*β*) and 95% confidence intervals (CI) were calculated using generalized estimating equations. The generalized estimating equation approach allows for shared variance between cohabitating and/or biologically related participants to be taken account using family groupings [43]. Independently, the parent E-DII scores were regressed onto each of the stress variables. Separately, C-DII scores were regressed onto the household chaos and family functioning variables. Parent and child models were separated due to different DII scoring protocols for adults and children. Parent models were adjusted for BMI, annual household income (5 categories: <$30,000; $30,000–59,999; $60,000–99,999; $100,000+; did not disclose), ethnicity (6 categories: Northeast or Southeast Asian; South Asian; White; mixed ethnicity; other ethnicity; did not disclose), education (3 categories: no postsecondary degree; postsecondary graduate; postgraduate training), sex and age. Child models were adjusted for BMI*z*-score, sex, and age. We also fit models that included interaction terms (e.g., chaos × sex) to examine whether the associations between these family stress factors and E-DII (or C-DII) differed by parent or child sex, respectively. We found no evidence of interaction by parent’s or child’s sex (results not shown); therefore, we report only results from the main effect model. *p* ≤ 0.05 was considered statistically significant for all tests.

## 3. Results

### 3.1. Family Characteristics

Table 1 displays the demographic characteristics of the parents and children. Among parent participants, the average age was approximately 35 years for mothers and 36 years for fathers. Most parent participants were White (mothers: 82%, fathers: 79%), married (mothers: 83%, fathers: 74%), university or college educated (mothers: 92%, fathers: 73%), and families had relatively high annual household income (nearly 50% earned $100,000 Canadian or more per year). Among child participants, the average age was 3.57 (±1.10) years for females and 3.76 (±1.11) years for males. 

### 3.2. E-DII and C-DII Scores

Table 2 displays the mean intakes for the nutrients included in the E-DII and C-DII calculations as well as mean scores. For adults, E-DII scores range from −9 (most anti-inflammatory) to +8 (most proinflammatory). The mean E-DII scores for mothers indicate a slightly anti-inflammatory dietary profile (−0.71 ± 0.15), whereas fathers’ mean E-DII was relatively neutral (−0.03 ± 0.17). For children, C-DII scores range from −8 (most anti-inflammatory) to +8 (most proinflammatory). Both girls’ and boys’ C-DII means indicated relatively neutral inflammatory profiles (0.04 ± 0.14 and −0.05 ± 0.15, respectively).

### 3.3. Family Stress Scores

Table 3 displays the number of participants in the high- and low-stress groups as well as the mean scores within the groups. Approximately 20% of the parent sample was classified as having high parenting distress or high depressive symptoms. Specifically, for parenting distress, parents in the high-distress group scored within the 90th percentile of the scale manual [28], compared to parents in the low-distress group who scored within the 57th percentile. Among parents and children, approximately 10% of the sample was classified as having high household chaos and high family dysfunction, respectively. Most (77–91%) families were considered “low stress”, as indicated by scores below clinical guidelines or the midpoint of the scale (Table 3). These results suggest that parents feel some strain associated with their role in the family and that families live in moderately busy, noisy, and/or disorganized homes but maintain good intrafamily cooperation and communication overall.

### 3.4. Linear Regression Results

Models were not stratified by sex after no statistically significant moderation by sex was detected for parents or children, as evidenced by nonsignificant sex and stress interaction terms (data not shown). Table 3 shows the results of the linear regressions, adjusted using generalized estimating equations to account for shared variance among family clusters. Compared to those with low household chaos (<37.5 points), parents with high household chaos had significantly higher E-DII scores (*β* = 0.973; 95% CI: 0.321, 1.624, *p* = 0.003), indicating a more proinflammatory dietary profile. Similarly, parents in dysfunctional families (≥2.17 points) had significantly higher E-DII scores compared to parents in well-functioning families (*β* = 0.967; 95% CI: 0.173, 1.761, *p* = 0.02). No significant associations were found between parents’ E-DII scores and parenting distress or depressive symptoms, nor were any associations found for C-DII scores.

## 4. Discussion

The purpose of this study was to investigate associations between the home environment and dietary patterns linked to inflammation among a sample of families with preschool-aged children in the Guelph Family Health Study. These results suggest that chaotic or dysfunctional family environments are associated with more proinflammatory dietary profiles for parents but not for children. No clinical guidelines for dietary inflammatory profiles exist; however, an anti-inflammatory profile is considered more health protective [37,38,39]. 

The majority of work on the DII and mental health among adults has posited that proinflammatory DII scores contribute to mental health status. For example, research has shown that low DII scores (indicating an inti-inflammatory diet) are associated with nearly 20% lower risk of depressive symptoms over time [6]. Additionally, greater DII scores (indicating a proinflammatory diet) were associated with greater likelihood of mood disorders and psychological distress in a number of cross-sectional studies [13]. These findings do support a cross-sectional association between parents’ proinflammatory dietary profiles and their levels of household chaos and family dysfunction; however, no significant associations were found for parenting distress or depressive symptoms. The current study frames the problem slightly differently from past research by suggesting that stressful home environments may serve as barriers to healthful food-related behaviours, and thus busy families may trend towards proinflammatory diet profiles. Additionally, more proinflammatory dietary profiles may impair resilience to stress, thus creating a vicious cycle. For parents, home chaos and family conflict have been linked with indices of poor diet quality such as lower fruit and vegetable consumption [44], which is consistent with the results of the present study. Although DII scores are not directly indicative of nutritional healthfulness, DII scores are inversely correlated with other diet quality indices such as the Healthy Eating Index [45] and positively correlated with glycaemic index scores [46]. Further, extensive work has demonstrated that higher DII scores pose an adverse risk on cardiometabolic health and chronic disease risk, including increased risk of cardiovascular disease [47,48], metabolic syndrome [48], obesity [49], and cancer [50]. Our finding that home chaos and family dysfunction are associated with greater inflammatory potential of the diet suggests that parent diet quality as well as chronic disease risk are of concern for stressed parents.

This study hypothesized that children in stressed families could have more proinflammatory dietary patterns. The suspected mechanisms driving this hypothesized association were children’s stress-eating and indirectly though parent-mediated behaviours such as fewer family meals among stressed families. Family meals are an important nutritional and social event for children; however, stressful home environments may be perceived by parents as barriers to meal preparation [51] and are inversely associated with family meal frequency [21,51] as well as healthful food availability in the home [51], which has an immediate spillover to children. Although past research has linked stressful home environments to poorer children’s diet quality [44] and elevated inflammatory markers in children [52], the results of the present study do not support a direct link between family stress and dietary sources of inflammation despite some significant associations found among parents.

As the C-DII is a recently developed tool, there is limited available research with which to compare our C-DII results. Early applications of the C-DII have focused on children’s risk for cardiometabolic disease and overweight/obesity. Most relevant to the present study is the Irish Lifeways Cross-Generation Cohort Study, which found that 5-year-old children in the highest tertile of C-DII scores had a 27% greater likelihood of being overweight or obese and had a 32% increased likelihood of obesity at age 9 compared to children in the lowest C-DII tertile [53]. Interestingly, Navarro and colleagues [53] found that children who consumed family meals at the table less than once a week had nearly twice the likelihood of having a higher C-DII score than children who ate family meals at the table every day; this finding is consistent with the family meals research discussed above. Hours spent watching TV was also positively associated with C-DII scores, which is in line with other research has linked higher family stress with less-regulated children’s screen time [25]. Taken together, these findings help to elucidate the potential mechanisms by which family stress may contribute to adverse health effects for children despite the lack of direct associations to dietary sources of inflammation found in the present study.

### Strengths and Limitations

This study contributes to the literature by examining the physiological consequences of chronic stress from a different perspective and in the under-researched context of family health. This study was able to explore associations between family stress and dietary inflammatory profile among both parents and children. Further, the inclusion of fathers rather than the typical mother–child dyad is another strength of this study. This comprehensive view of the family is paired with a multidimensional approach to stress measurement; examining multiple measures of stress better quantifies the many facets of an individual’s stress level. The E-DII and C-DII are extensively validated and literature-based tools that are able to powerfully conceptualize inflammation as an extension of dietary intake. These tools not only provide additional windows into inflammatory health, but they also circumvent the common barrier that blood sampling poses to participation, especially for children. 

Although this study adds novel insight into the DII and mental health literature, there are several limitations that should be considered when interpreting these results. To minimize participant burden in multiparent families, household chaos and family functioning were assessed only based on the primary parent to enroll in the study and those scores were extrapolated to the entire family. In this sample, the primary parents were predominantly mothers. Although some past work indicates similar family-based stress perception among mothers and fathers of young children [20], there is the potential for different family members to report family functioning differently [54]. This bias towards mothers’ perspectives is pervasive in the family health literature [55]. 

The E-DII and C-DII data can only be as accurate as the dietary intake assessment methodology. While practically advantageous for both researchers and participants, the self-administered nature of ASA24 may have limited the accuracy of the dietary intake assessment that must be considered in combination with the potential response bias of any nonsupervised reporting of dietary intake. This may be of particular concern where parents are reporting on children’s intake and may be deflecting criticism by reporting a more favourable diet than is actually being consumed, perhaps particularly among mothers [56]. Unfortunately, there is very little evidence about bias in proxy reporters. These concerns are tempered by ASA24’s reputation as a validated dietary assessment tool. An additional limitation is that ASA24 did not allow for the quantification of all 45 food parameters for optimal E-DII calculations. It is also possible that, despite adjustment for energy intake, the young children in this sample did not consume a sufficient quantity of food to allow us to detect underlying associations. Additionally, many factors, such as medical conditions, medications, injury, and environmental exposures, determine the circulatory levels of inflammatory markers. Although the E-DII and C-DII were designed to reflect inflammatory markers [37,38,39], there is the potential for a difference between individuals’ DII scores and their actual serum inflammatory profile. 

The relationship between stress and dietary intake is complex; it is conceivable that poor diet can negatively impact stress levels [57]. It is possible that testing multiple hypotheses may have increased the risk of type 1 error; however, the narrow confidence interval seen here supports the integrity of these findings. Further replication of this work, particularly among a more sociodemographically diverse sample than in the present study, is recommended. Additional work should also address this cross-sectional study’s inability to confirm any causal mechanisms underlying these results. Future research should also aim to investigate how physiological measures of chronic stress, such as hair cortisol concentrations, could increase our understanding of these stress and health dynamics.

## 5. Conclusions

The findings from the present study make an important contribution to our understanding of mental well-being and physical health in the family context. These results suggest that stressful home environments and family dynamics are associated with more proinflammatory dietary profiles among parents but not among preschool-aged children. Inflammatory pathways are implicated in a number of chronic health outcomes; as modifiable factors, a deeper understanding of the role that stress and dietary patterns may play in inflammation will help to inform health promotion strategies for families. 

## Figures and Tables

**Table 1 nutrients-13-01464-t001:** Family demographic characteristics (*n* = 611 participants from 241 families).

Characteristic	Mothers *n* = 212	Fathers *n* = 146	Girls *n* = 130	Boys *n* = 123
Age (years), mean (SE)	35.16 (0.32)	36.46 (0.40)	3.57 (0.10)	3.76 (0.11)
BMI (kg/m^2^), mean (SE)	26.80 (0.46)	26.93 (0.43)	0.48 (0.08) ^1^	0.54 (0.08) ^1^
Ethnicity, *n* (%)				
Northeast or Southeast Asian (e.g., Chinese, Korean, Japanese, Vietnamese, Cambodian, Filipino, Malaysian, Laotian, etc.)	11 (5.19)	6 (4.11)	4 (3.08)	5 (4.07)
South Asian (e.g., East Indian, Pakistani, Sri Lankan, etc.)	6 (2.83)	7 (4.79)	4 (3.08)	4 (3.25)
Other (e.g., Black, West Asian, Latin American)	15 (7.08)	9 (6.16)	6 (4.62)	3 (2.44)
White	174 (82.08)	115 (78.77)	103 (79.23)	88 (71.54)
Mixed ethnicity	5 (2.36)	7 (4.79)	10 (7.69)	18 (14.63)
Did not disclose	1 (0.47)	2 (1.37)	3 (2.31)	5 (4.07)
Education, *n* (%)				
No postsecondary degree	21 (9.91)	35 (23.97)	-	-
University or college graduate	105 (49.53)	71 (48.63)	-	-
Postgraduate training or degree	86 (40.57)	40 (27.40)	-	-
Annual household income, *n* (%)				
Less than $30,000	9 (4.25)	3 (2.05)	-	-
$30,000 to $59,999	24 (11.32)	24 (16.44)	-	-
$60,000 to $99,999	67 (31.60)	38 (26.03)	-	-
$100,000 or more	102 (48.11)	73 (50.00)	-	-
Did not disclose	10 (4.72)	8 (5.48)	-	-
Number of children, *n* (%)				
1	62 (29.95)	44 (28.77)	-	-
2	116 (54.72)	79 (54.11)	-	-
3 or more	34 (18.48)	25 (17.12)	-	-

^1^ BMI *z*-score, calculated per World Health Organization Child Growth Standards, adjusted for age and sex.

**Table 2 nutrients-13-01464-t002:** Summary of nutrient intakes used in E-DII and C-DII calculations and mean scores ^1^, stratified by sex. Intake assessed using the ASA24 self-reported online dietary recall program.

Nutrient	Mothers *n* = 212	Fathers *n* = 146	Girls *n* = 130	Boys *n* = 123
Total Energy (kcal)	2020.08 (48.86)	2540.31 (79.62)	1378.04 (33.41)	1481.98 (36.92)
Protein (g)	83.37 (2.33)	105.20 (3.83)	56.26 (1.76)	56.76 (1.99)
Carbohydrates (g)	234.99 (6.39)	284.04 (9.04)	175.62 (4.84)	197.04 (5.57)
Total fat (g)	84.92 (2.74)	102.69 (4.23)	52.98 (1.66)	54.97 (1.92)
Saturated fatty acids (g)	27.33 (1.01)	33.59 (1.61)	19.69 (0.79)	19.44 (0.80)
Monounsaturated fatty acids (g)	31.34 (1.10)	38.50 (1.62)	18.23 (0.60)	19.60 (0.74)
Polyunsaturated fatty acids (g)	18.85 (0.78)	21.42 (1.09)	10.04 (0.43)	10.86 (0.54)
Sodium (mg)	3222.27 (90.80)	4164.60 (148.83)	1992.24 (60.30)	2121.98 (71.45)
Alcohol (g)	3.27 (0.62)	13.94 (3.35)	-	-
Fibre (g)	22.68 (0.81)	22.56 (0.78)	14.62 (0.50)	15.63 (0.58)
Cholesterol (mg)	293.31 (14.20)	393.60 (27.81)	184.38 (9.93)	181.56 (11.88)
Niacin (mg)	22.16 (0.69)	30.62 (1.23)	13.16 (0.55)	13.90 (0.61)
Thiamin (mg)	1.84 (0.09)	2.44 (0.11)	1.17 (0.05)	1.32 (0.06)
Riboflavin (mg)	2.12 (0.05)	2.66 (0.10)	1.53 (0.05)	1.55 (0.05)
Vitamin B6 (mg)	1.79 (0.06)	2.32 (0.11)	1.16 (0.04)	1.18 (0.04)
Vitamin B12 (μg)	3.85 (0.18)	6.18 (1.18)	3.11 (0.17)	2.97 (0.16)
Iron (mg)	13.86 (0.41)	17.63 (0.68)	8.90 (0.33)	9.69 (0.36)
Magnesium (mg)	360.15 (10.05)	391.03 (13.58)	220.10 (7.51)	230.05 (7.55)
Zinc (mg)	11.44 (0.40)	14.49 (1.00)	7.67 (0.27)	7.60 (0.29)
Selenium (μg)	117.84 (4.08)	161.74 (8.25)	78.19 (2.69)	79.99 (2.93)
Vitamin A (μg)	879.52 (37.69)	881.96 (49.58)	622.99 (30.52)	600.20 (31.53)
Vitamin C (mg)	103.29 (5.85)	101.23 (7.05)	75.12 (4.53)	82.66 (6.15)
Vitamin E (mg)	11.27 (0.46)	12.98 (0.75)	5.98 (0.23)	6.57 (0.35)
Folic acid (μg)	110.36 (6.20)	170.79 (11.70)	79.30 (4.82)	95.84 (6.81)
Beta carotene (μg)	4845.70 (350.35)	3968.31 (389.87)	2282.77 (226.64)	2332.66 (233.88)
Dietary Inflammatory Index score ^1^, mean (SE)	−0.71 (0.15)	−0.03 (0.17)	0.04 (0.14)	−0.05 (0.15)
Anti-inflammatory profile (i.e., scores <0), *n* (%)	144 (67.9)	73 (50.0)	73 (56.1)	65 (52.9)
Proinflammatory profile (i.e., scores >0), *n* (%)	68 (32.1)	73 (50.0)	57 (43.9)	58 (47.2)

^1^ Adjusted for total energy intake. Separate adult and child scoring protocols. Data are means (SE) unless otherwise indicated.

**Table 3 nutrients-13-01464-t003:** Associations between family stress levels and Dietary Inflammatory Index scores ^1^ among preschool-aged children and their parents (*n* = 241 families). Results presented as means (SE) with linear regression coefficient estimates (*β*) using generalized estimating equations with 95% confidence intervals (CI) and *p*-values (significant findings shown in bold).

Stress Measure	High Stress Mean (SE) *n* (%)	Low Stress Mean (SE) *n* (%)	β (95% CI) *p*-Value
**Parents**			
Parenting Distress (High ≥ 36 points)	39.56 (0.34) *n* = 68 (19.0%)	26.48 (0.33) *n* = 290 (81.0%)	−0.107 (−0.713, 0.498) *p* = 0.73
Depressive Symptoms ^2^ (High ≥ 10 points)	13.64 (0.42) *n* = 83 (23.3%)	4.30 (0.15) *n* = 273 (76.7%)	0.234 (−0.329, 0.797) *p* = 0.42
Household Chaos ^3^ (High ≥ 37.5 points)	40.93 (0.47) *n* = 40 (11.4%)	28.43 (0.26) *n* = 310 (88.6%)	**0.973 (0.321, 1.624)** ***p* = 0.003**
Family Dysfunction ^4^ (High ≥ 2.17 points)	2.42 (0.0.04) *n* = 31 (8.7%)	1.55 (0.0.02) *n* = 324 (91.3%)	**0.967 (0.173, 1.761)** ***p* = 0.02**
**Children**			
Household Chaos ^3^ (High ≥ 37.5 points)	40.73 (0.49) *n* = 30 (12.1%)	28.88 (0.32) *n* = 219 (88.0%)	0.221 (−0.450, 0.893) *p* = 0.52
Family Dysfunction ^4^ (High ≥ 2.17 points)	2.39 (0.04) *n* = 24 (9.6%)	1.57 (0.02) *n* = 226 (90.4%)	0.036 (−0.624, 0.696) *p* = 0.92

^1^ Parent models adjusted for body mass index (BMI), annual household income (<$30,000; $30,000–59,999; $60,000–99,999; $100,000+; did not disclose), ethnicity (Northeast or Southeast Asian; South Asian; White; mixed ethnicity; other ethnicity; did not disclose), education (no postsecondary degree; postsecondary graduate; postgraduate training), sex, and age. DII calculations adjust for total energy intake. Child models adjusted for BMI*z*-score, sex, and age. Child DII scoring protocol used, which includes adjustment for total energy intake. ^2^ Two parents did not complete all items of the depression survey, *n* = 356 parents. ^3^ Five families did not complete all items of the chaos survey, *n* = 350 parents and 250 children. ^4^ Four families did not complete all items of the family functioning survey, *n* = 355 parents and 250 children.

## Data Availability

The Guelph Family Health Study (GFHS) welcomes outside collaborators. Interested investigators can contact GFHS investigators to explore this option, which preserves participant confidentiality and meets the requirements of our Research Ethics Board, to protect human subjects. Due to Research Ethics Board restrictions and participant confidentiality, we do not make participant data publicly available.

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
