# Peer review of "Associations between Family-Based Stress and Dietary Inflammatory Potential among Families with Preschool-Aged Children"

_nutrients, 2021, doi:10.3390/nu13051464_

Round 1

Reviewer 1 Report

Thank you for the opportunity to review this manuscript presenting interesting and innovative research. The study uses an existing cohort to investigate the association between family-based stress and diet, with potential implications for a broad range of community and public health strategies.

The introduction is very well written in clear language, citing relevant literature and clearly outlining a rationale for this research.

The methods provide a highly detailed description of the methods used. All methods used are appropriate and demonstrate a good understanding of the field of research and analytic techniques.

In section 3.2, results very near to 0 (neither pro- nor anti-inflammatory) have been reported a slightly pro- or anti-inflammatory. While this would seem appropriate for the mothers, I feel it would be more appropriate to categorise the scores for fathers, girls and boys as neutral (neither pro- nor anti-inflammatory).

All results are otherwise presented clearly and concisely. Tables are appropriate and used sparingly.

The discussion critically evaluates the findings in light of other relevant literature and limitations in the methods used. It provides relevant conclusions and recommendations for future research.

Reviewer 2 Report

The purpose of the study was to investigate the association between family-based stress and the inflammatory potential of the diet among preschool-aged children and their parents.

Major comment/concern

The fact that the authors were not able to validate their theory could be due to the limitation associated with C-DII.  The authors used C- DII on children of average age of 3 years. About 20 of the children seemed to be of a very small age (weighed in their parents’ arm according to the method section). This tool was validated in children aged 6-14 years. This may explain the lack of association with their parents E-DII scores. Because small children’s meals are lower in volume, and may vary for younger age groups, this could have been the limitation and should be noted in the manuscript.

Minor comments

  1. The title is a misleading by starting with “brain food”. I suggest that the authors drop it to give more emphasis on the actual goal of the study.
  2. Please revise second sentence, it sounds incoherent.
  3. Fiber throughout the manuscript and in table 2 is written the French way. Please revise.
  4. What was the length of the study? I am thinking about the seasonal implication on diet quality and mood. Could this variable have played a factor is skewing the results?
  5. Did the author assess the stress level of the pre-school children? This would be an indirect way to validate the C-DII results. As the author mentioned in the discussion, mostly mothers completed the survey. Mothers’ perspective could be biased based on the perception of being overwhelmed, not getting enough help, piling tasks, etc.., due to giving their children the nurture needed (which will lead to a low stress in children) or because of feeling overwhelmed they are likely to neglect some aspects of nurture (which may cause higher stress in children).
  6. Line 349, this association is bidirectional. A low diet quality lowers resilience to stress, which becomes a vicious cycle.
  7. Line 370, please provide reference for Navarro et al.
  8. A comma and a semi-colon should precede “which” and “however”, respectively. There is a need to revise throughout the manuscript.
  9. Line 355, a very long sentence, please revise.
  10. It is better to divide the last part of the manuscript into strengths and limitations section and conclusion.

Round 2

Reviewer 2 Report

The authors have done a good job reviewing the manuscript. They have addressed all my comments and concerns.